# Risk of Mortality from Respiratory Malignant and Non-Malignant Diseases among Talc Miners and Millers: A Systematic Review and Meta-Analysis

**DOI:** 10.3390/toxics10100589

**Published:** 2022-10-05

**Authors:** Catalina Ciocan, Alessandro Godono, Sandro Stefanin, Paolo Boffetta, Enrico Pira, Marco Clari

**Affiliations:** 1Department of Public Health and Pediatrics, University of Torino, 10126 Turin, Italy; 2Pole of the CTO Hospital Unit of the “Ferdinando Rossi” Federal Library of Medicine, 10126 Turin, Italy; 3Department of Medical and Surgical Sciences, University of Bologna, 40126 Bologna, Italy; 4Stony Brook Cancer Center, Stony Brook University, Stony Brook, NY 11794, USA

**Keywords:** mortality, talc mining industry, respiratory diseases

## Abstract

There is contrasting data on the association between talc exposure and lung and pleural cancer. Given the potential importance of this aspect, we performed a systematic review and meta-analysis to investigate the association between working in the talc extractive industry and mortality from malignant and non-malignant respiratory diseases. We followed PRISMA guidelines to systematically search for pertinent articles in three relevant electronic databases: Pubmed, Scopus, and WebOfScience, from their inception to 30 November 2021. The methodological quality of included articles was evaluated using the US National Institutes of Health tool. Standardized incidence ratios (SIRs) and standardized mortality ratios (SMRs) for malignant and non-malignant respiratory diseases as well as respective 95% confidence intervals (CIs) were extracted or calculated for each included cohort. Six articles comprising 7 cohorts were included in the metanalysis. There was increased mortality for pneumoconiosis, especially in the miner’s group (SMR = 7.90, CI 95% 2.77–22.58) and especially in those exposed to higher quartz concentration and for non-malignant respiratory diseases in the overall analysis (SMR = 1.81, CI 95% 1.15–2.82). The risk for lung cancer mortality was slightly increased in the overall analysis (SMR = 1.42, CI 95% 1.07–1.89). The risk for malignant mesothelioma could not be calculated due to an insufficient number of studies assessing this outcome. This systematic review and meta-analysis provides evidence that men working in the talc mining industry have increased mortality for non-malignant respiratory diseases including pneumoconiosis. The small excess in lung cancer mortality may be, in part, explained by the high prevalence of the smokers in some of the analyzed cohorts or by the exposure to other carcinogens like radon decay products and diesel engine exhaust.

## 1. Introduction

Talc is a natural lamellar structured silicate, widely used in industrial and commercial products as well as for cosmetic and therapeutic purposes. Talc containing asbestos (“fibrous talc”) was first classified in 1997 [1] and confirmed in 2010 [2] by the International Agency for Research on Cancer (IARC) as a group 1 agent (carcinogenic to humans) while talc with no detectable level of asbestos was classified as a group 3 agent (unclassifiable as to carcinogenicity in humans).

Occupational exposure to talc dust occurs during its mining, crushing, separating, bagging, and loading, and in various industries that use talc [1].

Regarding non-malignant respiratory diseases, talc pneumoconiosis was described for the first time by Thorel in 1896 [3]. Since then, other studies have shown an association between pneumoconiosis and asbestos-containing talc [4,5,6] as well as talc mixed with silica, another potential contaminant of talc [7,8,9,10,11,12,13].

Concerning possible carcinogenic effects on the lung and pleura, several studies of workers exposed to talc with no detectable level of asbestos reported no cases of mesothelioma and no excess risk of lung cancer [7,8,9,10,11,14,15].

Mesothelioma cases have been reported in workers of talc contaminated with asbestos, and a meta-analysis of 14 cohort studies, including cohorts of workers exposed to asbestos-contaminated talc, showed a slight increase in lung cancer risk among exposed subjects [16] with no difference according to potential asbestos contamination. Other authors observed occasional excesses of lung cancer risk among miners, not paralleled by comparable results in millers, attributed to co-exposure to radon decay products [15] or crystalline silica [17].

Due to the limited evidence on talc exposure and mortality for lung cancer, malignant mesothelioma, pneumoconiosis, and non-malignant respiratory diseases, we aimed at performing a systematic review and meta-analysis to investigate the risk of mortality from respiratory malignant and non-malignant diseases among talc miners and millers.

## 2. Materials and Methods

This systematic review and meta-analysis was performed and reported according to the Preferred Reporting Items for Systematic Reviews and Meta-Analyses (PRISMA) guidelines [18].

### 2.1. Search Strategy

We conducted a systematic search for relevant articles in three electronic databases: Pubmed, Scopus, and WebOfScience, from their inception to 30 November 2021. No limits were applied for language, and results were limited to studies conducted on workers in the talc extractive industry. A comprehensive search strategy was applied using the following terms: “Miners”, “Millers”, “Talc”, “Mortality”, “Incidence”, “Cancer”, “Tumor”, “Neoplasm”, “Carcinoma”, “Respiratory”, “Disease”, “Pneumoconiosis”, and “Silica”. Databases searches were conducted with the aid of an expert librarian to ensure rigor. The search strategies for each database are illustrated in Appendix A.

### 2.2. Inclusion and Exclusion Criteria

We included articles reporting the results of cohort studies and case-control studies nested in cohorts of workers employed in the extractive talc industry in which respiratory occupational talc exposure was considered substantial.

We included studies reporting results on incidence and mortality from malignant and non-malignant respiratory effects derived from talc mining; results on respiratory function were excluded. Only full articles published in peer-reviewed journals were considered. Descriptive studies, as well as case reports, conference proceedings, theses, letters to the editor, and articles for which the full text was not available either online or by request to the journal, were excluded. Other systematic reviews or meta-analyses were included in the review but not in the meta-analysis.

Two independent reviewers assessed the articles based on the inclusion and exclusion criteria. In case of disagreement, a third reviewer’s evaluation was requested.

### 2.3. Data Extraction

The outcome measures extracted for each paper were the Standardized Incidence Ratio (SIR) and Standardized Moltality Ratio (SMR) for lung cancer, malignant mesothelioma, pneumoconiosis, and other non-malignant respiratory diseases (NMRD), as well as their 95% confidence intervals (CIs). When articles provided the number of observed and expected cases for one of the 4 outcomes but no SIRs or SMRs, these measures and their respective 95% CIs were calculated. In the case of multiple reports from the same cohort, the most complete results (i.e., those based on the largest number of cases) were used.

Other data were also extracted when reported in the article: year of publication, study design, country, sample size, type of job, number of person-years, duration of employment, duration of follow-up, minimum time of exposure, total dust exposure levels, crystalline silica levels (reports as a percentage of total dust), the presence of asbestos, the prevalence of smokers, and outcome (incidence or mortality). We also abstracted data on the exposure levels to total dust and free silica. Data were extracted from two independent reviewers, and any disagreement was solved with a third reviewer.

### 2.4. Quality Assessment

We assessed the methodological quality of the included articles by choosing the quality assessment tool of the US National Institutes of Health, based on the study design [19]. The methodological quality of the articles was to be rated as poor (score = 7–8), fair (score = 9), and good (score = 10–12). Only articles with a quality score ≥ 9 were included.

### 2.5. Meta-Analysis Statistical Method

The results of the included studies were combined for lung cancer, malignant mesothelioma, pneumoconiosis, and non-malignant respiratory diseases, using random-effects meta-analyses [20] based on the log-transformation of the SMR/SIR and its standard error. The inter-study heterogeneity was assessed by the I2 test [21].

Stratified meta-analyses were conducted by type of job (miners and millers). The presence of publication bias was assessed by visual inspection of the funnel plots and calculating the Egger test [22].

## 3. Results

The search of the PubMed, Scopus, and WebOfScience databases yielded a total of 289 articles (93 in Scopus, 133 in Pubmed, and 63 in WebOfScience). 148 articles remained after duplicates were removed. After reviewing the titles and the abstracts, 21 articles were considered possibly relevant, and of these, 3 articles were discarded following a review of the full text because they did not meet the inclusion criteria.

The full texts of the remaining 18 articles were examined in detail. 12 articles represented early reports of studies which were later updated. Thus, 6 articles met the inclusion criteria for the systematic review (Figure 1).

A manual search of the reference lists of included articles revealed 2 additional relevant studies [23,24] of which one [23] was updated by a later study already included in the meta-analysis [13] and the other was not available in full-text version [24].

The quality score of all included articles was good, ranging from 10 to 12.

The included articles reported results from a total of 7 cohorts of men with talc exposure in the extractive industry. Included articles were published from 1980 to 2022. Four of the cohorts were studied in Europe (Italy, Norway, France, and Austria) and the remaining 3 in the USA (2 from New York State and 1 from Vermont). All studies were of cohort design, except that by Wild et al., which consisted of a case-control study nested in a cohort. All studies evaluated mortality among talc miners and millers.

A total of 5394 miners and millers were considered, with a follow-up duration ranging from 28 to 74 years. They were all males. The number of workers in each cohort ranged from 390 [15] to 1749 [11]. Mortality data reported in the articles were mainly retrieved from death certificates and in one study [15] from cancer registries. Minimum employment time was reported for all cohorts and ranged from 1 day to 1 year.

Talc from one mine from Vermont contained tremolite and anthophyllite, while talc from the Norwegian cohort contained traces of asbestos.

Quartz was present in small amounts (<1%) in basal samplings of the total dust, in all cohorts, except for one (Norwegian cohort) that found silica only in traces. During drilling operations, silica varied from <1% to 16% of the total dust (Table 1).

The prevalence of smokers varied between 42% to 73% in the analyzed group samples (Table 2).

Selected characteristics of the cohorts are reported in Table 3. Results of individual studies are summarized in Appendix A.

### Meta-Analysis

The results of the meta-analysis showed increased mortality for pneumoconiosis (Table 4), especially in the miner’s group (SMR = 7.90, CI 95% 2.77–22.58) (Table 5). Mortality for pneumoconiosis was associated with quartz concentration (quartz concentration > 1%, SMR = 9.55, CI 95% 7.48–12.19 vs. quartz concentration < 1%, SMR = 5.82, CI 95% 3.11–10.91) (Table 4).

The risk for lung cancer mortality was slightly increased in the overall analysis (SMR = 1.42, CI 95% 1.07–1.89). There was no dose-response relation with quartz concentration (quartz concentration > 1%, SMR = 1.56, CI 95% 0.60–4.02 vs. quartz concentration < 1%, SMR = 1.27, CI 95% 1.01–1.60)

Mortality for non-malignant respiratory diseases in the overall analysis was increased (SMR = 1.81, CI 95% 1.15–2.82).

The risk for malignant mesothelioma could not be calculated due to an insufficient number of studies assessing this outcome. Only one study (Fordyce 2019 [12]) reported the SMR = 1.004, with one mesothelioma death after the reanalysis of the death certificates by a trained nosologist. Another death for mesothelioma was reported by Lamm and Starr [27] in an earlier follow-up of this cohort, but was not confirmed by Fordyce et al. There were no cases of mesothelioma reported in the cohorts from Italy [11], France, and Austria [13]. Dement et al. [25] reported one death due to mesothelioma in the study population. Honda et al. [26] found two deaths from mesothelioma; these authors stated that because of the short latency for the first case and the low talc exposure of the second case, it is unlikely that either case was due to talc ore dust. Overall, there were five deaths for mesothelioma in all the cohorts.

The forest plots of the meta-analyses for pneumoconiosis, non-malignant respiratory diseases and lung cancer are represented, respectively, in Figure 2, Figure 3 and Figure 4.

The visual inspection of the funnel plots revealed no publication bias for lung cancer, pneumoconiosis, and non-malignant respiratory diseases meta-analyses (Appendix A). Moreover, no small study effect was identified (*p* > 0.05 for all outcomes).

## 4. Discussion

Overall, our study shows evidence that occupational exposure in the talc mining industry is positively related to increased mortality from pneumoconiosis, lung cancer, and non-malignant respiratory diseases.

Mortality for pneumoconiosis was higher among miners and was associated with higher quartz concentrations. This result confirms previous observation from the Italian, French, and American cohorts, while there was no excess of pneumoconiosis in the Norwegian and Austrian cohorts.

The risk for lung cancer mortality was slightly increased and this may be, in part, explained by the high prevalence of the smokers in some of the analyzed cohorts. Another explanation may be the exposure to other carcinogens like radon decay products and diesel engine exhaust. There was no dose-response relation with quartz concentration, meaning that silica is not a plausible cause for the excess in lung cancer mortality.

Mortality for non-malignant respiratory diseases also increased and was associated with quartz concentration and possibly with total dust levels. However, in this meta-analysis, we evaluated the association between talc exposure and the occurrence of non-malignant respiratory diseases including those with a high incidence in the general population such as asthma and COPD. These diseases recognize multiple etiological factors, both environmental (cigarette smoke, air pollution, allergens) and individual (genetic susceptibility, autoimmunity, etc.). In particular, recent studies highlighted the role of epigenetic mechanism in the pathogenesis of asthma, allergic rhinitis, and other allergic disorders, especially through mediating the effects of the environmental factors, well recognized risk modifiers [28]. To our knowledge, studies that have investigated possible epigenetically based pathogenetic mechanisms of talc-related respiratory diseases are very limited [29] and future research is needed to further investigate this possible effect.

The risk for malignant mesothelioma could not be calculated due to an insufficient number of studies assessing this outcome. However, the available data do not suggest an increased risk of this neoplasm among talc miners and millers [30].

This systematic review and meta-analysis have several strengths, such as the broad search strategy, aiming at including all studies reporting results regarding talc occupational exposure.

An important aspect of this review is the assessment of talc and silica exposure as well as the contamination by asbestiform fibers.

Unlike other studies that have included both extractive industries and industries that use talc for their end-products, this meta-analysis included only talc mining cohorts, and this aspect allows us to observe health effects in cohorts with comparable occupational conditions.

Furthermore, all studies had a good methodological quality (score between 10–12).

Our systematic review and meta-analysis also suffer from some limitations. The results are based on a small number of studies as there are only a few cohorts of talc miners and millers worldwide. In addition, our results could be influenced by the high prevalence of smokers in the cohorts. Data on smoking prevalence were available on a reduced number of workers, thus data could not be sufficiently reliable to assess whether smoke influences mortality from respiratory malignant and non-malignant diseases.

In conclusion, the present study supports a positive association between talc mining and milling and mortality for pneumoconiosis, and non-malignant respiratory diseases. The risk for lung cancer mortality may be, in part, explained by the high prevalence of the smokers in some of the analyzed cohorts and by the exposure to other carcinogens like radon decay products and diesel engine exhaust. Results on mesothelioma are limited but they do not suggest an excess risk. The safety and the exposure of talc miners and millers to occupational agents should be carefully assessed and corrected by improving ventilation techniques and drilling procedures.

## Figures and Tables

**Figure 1 toxics-10-00589-f001:**
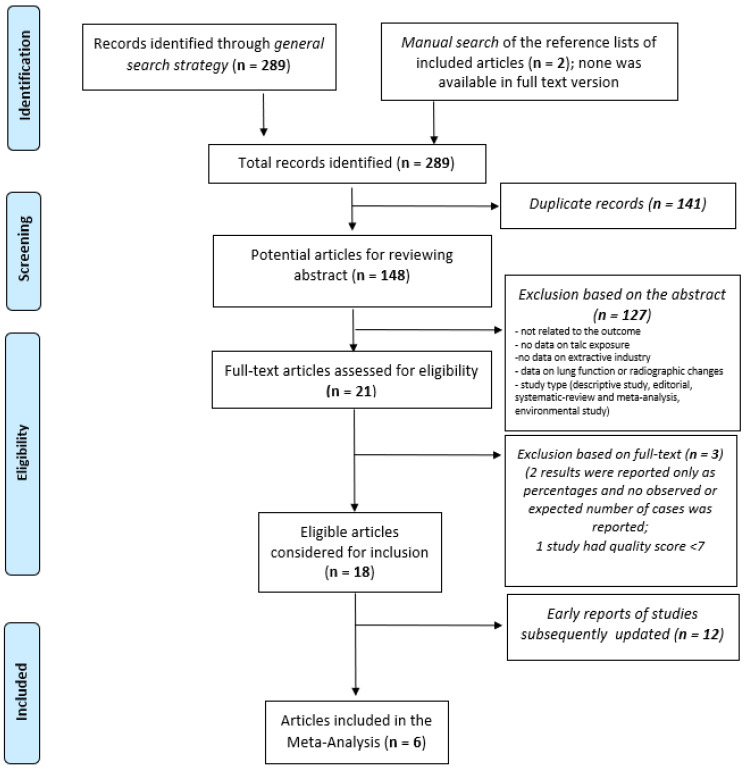
Flowchart of the study selection for the meta-analysis based on the inclusion and exclusion criteria.

**Figure 2 toxics-10-00589-f002:**
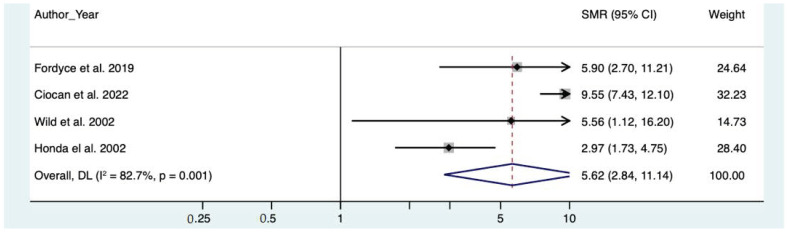
Meta analysis of pneumoconiosis [7,11,12,26].

**Figure 3 toxics-10-00589-f003:**
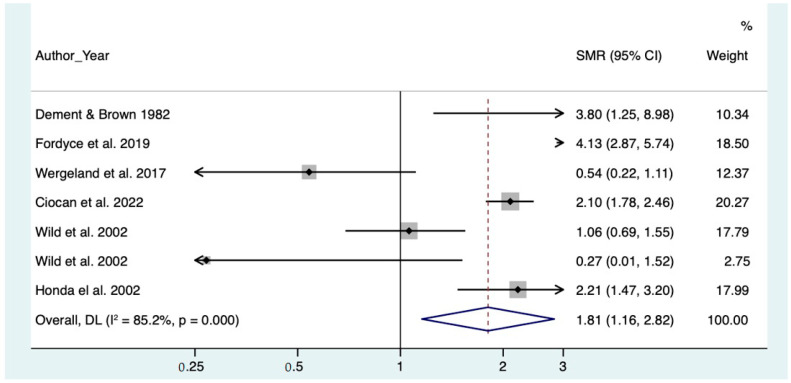
Meta analysis of non-malignant respiratory diseases [11,12,13,15,25,26].

**Figure 4 toxics-10-00589-f004:**
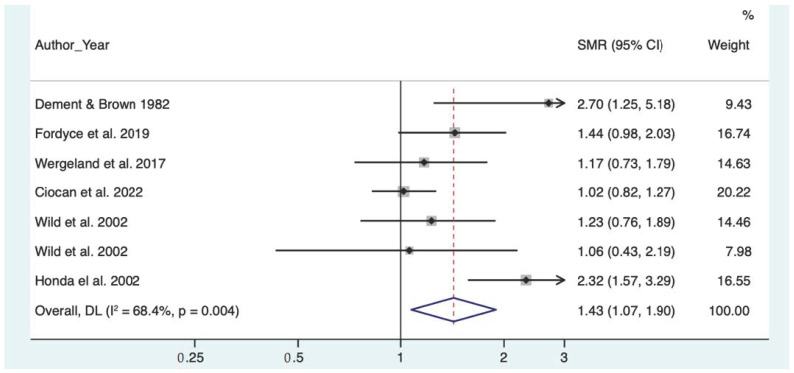
Meta analysis of lung cancer [11,12,13,15,25,26].

**Table 1 toxics-10-00589-t001:** Cohort exposure levels to total dust, silica, asbestos.

Author(s), Year	Total Dust (mppcf or mg/m^3^) and Time Period	Free Silica Exposure Level	Presence of Asbestos
Dement J.M. et al., 1980 [25]	0.25–2.96 mg/m^3^ (time period not stated)	0.04 mg/m^3^	fibrous tremolite and anthophyllite
Fordyce T.A. et al., 2019 [12]	not available	<1	no
Wergeland E. et al., 1990 [15]	0.94–318.9 mg/m^3^ (1980–1982)	3–6% during drilling operations (after 1981)	only trace amounts of tremolite and anthophyllite
Ciocan C. et al., 2022 [11]	0.8–790 mppcf (before 1965) <10 mppcf (1965–2003) 0.5–2.5 mg/m^3^ (after 2003)	10% of total dust (before 1974) 2% of total dust (1978) <1% (after 2003)	No
Honda et al., 2002 [26]	0.01–2.67 mg/m^3^	not stated	no
Wild P. et al., 2002 (France) [13]	0.5 mg/m^3^–15 mg/m^3^ (1986–1991)	<3% of total dust	no
Wild P. et al., 2002 (Austria) [13]	6.14 mg/m^3^	0.5–4% of total dust	0.25 fibers/cm^3^ (in the PM < 10 µm sample) 0.12 fibers/cm^3^ (in the PM < 2 µm sample) Analyses by polarized light microscopy indicated the presence of asbestiform tremolite, chrysotile, and anthophyllite in these samples.

**Table 2 toxics-10-00589-t002:** Smokers’ prevalence.

Author(s), Year	Smokers’ Prevalence (%)	Sample Size and Smokers Prevalence
Dement J.M. et al., 1980 [25]	48–79%	121 subjects, 48% smokers and 31% ex smokers (data from Dement, 1982)
Fordyce T.A. et al., 2019 [12]	64–99%	22 cases of lung cancer, 91% smokers and 9% ex smokers 66 controls, 64% smokers and 9% ex smokers (data from Gamble 1993)
Wergeland E. et al., 1990 [15]	70–76%	94 miners, 76% smokers 296 millers, 70% smokers
Ciocan C. et al., 2022 [11]	45–51%	in 1993, 200 workers, 45% smokers in 2010, 52 workers, 51% smokers (data from Pira, 2017)
Honda et al., 2002 [26]	73%	818 workers, 73% smokers
Wild P. et al., 2002 [13]	42–59%	Austrian cohort: 542 workers, in 1988, 42% smokers French cohort: 1070 workers, in 1989, 59% smokers

**Table 3 toxics-10-00589-t003:** Included cohorts’ characteristics.

Author(s), Year	Country	Study Design	Follow-Up Period	The First Year of Employment	Presence of Asbestos	No. of Subjects	Diagnostic Evidence	Outcome Studied	Quality Assessment
Dement J.M. et al., 1980 [25]	US	cohort	28	1947	Yes	398	Death certificate	Mortality	11
Fordyce T.A. et al., 2019 [12]	US	cohort	72	1930	No	200	Death certificate	Mortality	10
Wergeland E. et al., 1990 [15]	Norway	cohort	58	1953	only in traces	390	Cancer register	Mortality	11
Ciocan C. et al., 2022 [11]	Italy	cohort	74	1946	No	1749	Death certificate	Mortality	12
Honda et al., 2002 [26]	US	cohort	39	1948	No	809	Death certificate	Mortality	12
Wild P. et al., 2002 [13]	Austria	cohort-nested case-control	50	1972	No	542	Death certificate	Mortality	11

**Table 4 toxics-10-00589-t004:** SMR pooled results for malignant and non-malignant respiratory diseases.

Subgroup Items	SMR Pooled Results (95% CI) Non-Malignant Lung Diseases	SMR Pooled Results (95% CI) Malignant Lung Diseases	SMR Pooled Results (95% CI) Malignant Pleural Diseases	SMR Pooled Results (95% CI) Pneumoconiosis
all cohort	1.80 (1.15–2.82)	1.42 (1.07–1.89)	insufficient data to perform analysis	5.62 (2.83–11.14)
silica > 1%	2.29 (1.51–3.47)	1.56 (0.60–4.02)	-	9.55 (7.48–12.18)
silica < 1%	1.13 (0.37–3.42)	1.27 (1.01–1.60)	-	5.82 (3.10–10.91)

SMR: Standardized Mortality Ratio; CI: Confidence Interval.

**Table 5 toxics-10-00589-t005:** SMR pooled results for all respiratory diseases for miners and millers categories.

Subgroup Items	SMR Pooled Results (95% CI) Non-Malignant Lung Diseases	SMR Pooled Results (95% CI) Malignant Lung Diseases	SMR Pooled Results (95% CI) Malignant Pleural Diseases	SMR Pooled Results (95% CI) Pneumoconiosis
Miners	2.82 (2.02–3.95)	1.55 (0.75–3.19)	1.00 (0.18–5.51)	7.90 (2.76–22.58)
Millers	1.52 (0.69–3.35)	1.18 (0.91–1.52)	-	2.64 (1.40–4.96)

SMR: Standardized Mortality Ratio; CI: Confidence Interval.

## Data Availability

Not applicable.

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
