# Peer review of "Risk of Mortality from Respiratory Malignant and Non-Malignant Diseases among Talc Miners and Millers: A Systematic Review and Meta-Analysis"

_toxics, 2022, doi:10.3390/toxics10100589_

Round 1
Reviewer 1 Report
In this manuscript, Catalina Ciocan et al. provide evidence in a systematic review and meta-analysis that men working in talc mining industry have increased mortality for non-malignant respiratory diseases, including pneumoconiosis, according to the Preferred Reporting Items for Systematic Reviews and Meta-Analyses (PRISMA) guidelines.
The evidence shown here is very interesting and relevant. However, the way it is presented should be improved. Therefore, we take the task of making some suggestions.
Minor points
Authors should adequately list the tables by occurrence,
For example, in line 145. During drilling operations, silica varied from < 1% to 16% of the total dust (Table 4). It should follow table 2.
It does not mention table 5 in the text.
Figures 1, 2, 3, and 4 are not referenced in the text.
Authors should expand the discussion.
Author Response
Dear reviewer, thank you for your suggestions that can greatly improve the overall quality of our work.
- Authors should adequately list the tables by occurrence. We modified the tables' order and changed the captions.
-
Figures 1, 2, 3, and 4 are not referenced in the text. We added the missing references.
-
Authors should expand the discussion. We have expanded the discussion section by focusing on epigenetic aspects.

Reviewer 2 Report
Actually, this article is almost perfectly fine as it is now. Just briefly, the idea behind is good including the topic and scientific questions, the design is correct as based on the commonly accepted PRISMA guideliness, the data are nicely converted into results, the results are also nicely interpreted and analyzed in the discussion.
One small technical comment: please, try to improve the graphical quality of the figures (resolution).
A facultative comment: your research refers to toxic effects of the environmental influences. However, environment could affect our health also through other mechanisms, eespecially epigenetics. The role of epigenetics in mediating the environmental effects on respiratory diseases such as asthma, COPD and others lung diseases could be mentioned in 1-2 sentences in the discussion (PMID: 31633569, 35681424, 35008971).
Author Response
Dear reviewer,
thank you for your comments and appreciation about our study. We think that it's a relevant topic, worthy of further investigations.
-
One small technical comment: please, try to improve the graphical quality of the figures (resolution). We substituted/modified most of the figures included.
- A facultative comment: your research refers to toxic effects of the environmental influences. However, environment could affect our health also through other mechanisms, eespecially epigenetics. The role of epigenetics in mediating the environmental effects on respiratory diseases such as asthma, COPD and others lung diseases could be mentioned in 1-2 sentences in the discussion (PMID: 31633569, 35681424, 35008971). Thank you for your suggestions, we have expanded the discussion section by focusing on epigenetic aspects.
